# Examining the Influencing Factors of Chronic Hepatitis B Monitoring Behaviors among Asian Americans: Application of the Information-Motivation-Behavioral Model

**DOI:** 10.3390/ijerph19084642

**Published:** 2022-04-12

**Authors:** Grace X. Ma, Lin Zhu, Wenyue Lu, Yin Tan, Jade Truehart, Cicely Johnson, Elizabeth Handorf, Minhhuyen T. Nguyen, Ming-Chin Yeh, Min Qi Wang

**Affiliations:** 1Center for Asian Health, Lewis Katz School of Medicine, Temple University, Philadelphia, PA 19140, USA; lin.zhu@temple.edu (L.Z.); wenyue.lu@temple.edu (W.L.); ytan@temple.edu (Y.T.); jade.truehart@temple.edu (J.T.); 2Department of Urban Health and Population Science, Lewis Katz School of Medicine, Temple University, Philadelphia, PA 19140, USA; 3Department of Sociology, College of Liberal Arts, Temple University, Philadelphia, PA 19120, USA; 4Center for Cancer Health Disparities Research (CCHDR), Hunter College, City University of New York, New York, NY 10065, USA; cjpolicydr@gmail.com; 5Fox Chase Cancer Center, Temple University Health System, Philadelphia, PA 19111, USA; elizabeth.handorf@temple.edu (E.H.); minhhuyen.nguyen@fccc.edu (M.T.N.); 6Nutrition Program, Hunter College, City University of New York, New York, NY 10065, USA; myeh@hunter.cuny.edu; 7School of Public Health, University of Maryland, College Park, MD 20742, USA; mqw@umd.edu

**Keywords:** health disparities, cancer prevention, Asian Americans, chronic hepatitis B, monitoring adherence

## Abstract

Background: Compared to non-Hispanic whites, Asian Americans are 60% more likely to die from the disease. Doctor visitation for chronic hepatitis B (CHB) infection every six months is an effective approach to preventing liver cancer. Methods: This study utilized baseline data from an ongoing randomized controlled clinical trial aimed at improving long-term adherence to CHB monitoring/treatment. Guided by the information-motivation-behavioral skills (IMB) model, we examined factors associated with CHB monitoring adherence among Asian Americans with CHB. Multivariable logistic regression was conducted to test the associations. Results: The analysis sample consisted of 382 participants. Multivariable logistic regression showed that HBV knowledge (OR = 1.24, *p* < 0.01) and CHB-management motivation (OR = 1.06, *p* < 0.05) are significant predictors of having a doctor’s visit in the past six months. Both factors were positively associated with the likelihood of having had blood tests for HBV in the past six months. Conclusion: We found that greater HBV-related knowledge and CHB-management motivation are significantly associated with performing CHB-monitoring behaviors in the past six months. The findings have critical implications for the development and implementation of evidence-based interventions for CHB monitoring and liver cancer prevention in the Asian American community.

## 1. Introduction

Asian Americans are disproportionately affected by liver cancer and viral hepatitis [1,2,3]. The incidence of liver and intrahepatic bile duct (IBD) cancer is much higher among Asian Americans than in non-Hispanic whites (19.9 versus 10.8 per 100,000 men, and 7.4 versus 3.7 per 100,000 women) [4]. Asian Americans are also 70% more likely to die from liver and IBD cancer than their non-Hispanic white counterparts [4]. One major contributor to such alarming disparities is chronic hepatitis B (CHB) infection. Hepatitis B virus (HBV) infection is the leading cause of cirrhosis, and hepatocellular carcinoma (HCC) [2,5,6,7,8,9,10,11,12]. About 2.2 million individuals in the U.S. are living with CHB infection. Up to 70% of infected individuals are from non-U.S.-born populations, with the highest prevalence among persons from Asia (58%) [12,13,14,15,16,17,18,19,20,21,22,23]. Prevalence rates of CHB in Asian Americans range from 8% to 13%, a figure that is significantly higher than the 1% rate observed in non-Hispanic Whites (NHW) [1,22].

Regular monitoring for CHB is critical for tracking disease progression, treatment adherence, and risk for the development of HCC [24]. Guidelines from the American Association for the Study of Liver Disease (AASLD) recommend regular monitoring of individuals living with HBV at six-month intervals, with an assessment of liver function, HBV viral load, ultrasound, and alpha-fetoprotein (AFP) levels [25]. Routine monitoring for liver disease activity can inform treatment decisions aimed at reducing HCC risk [25]. Unfortunately, adherence to HCC surveillance has been suboptimal among Asian Americans, with less than 50% of those who are eligible to undergo regular CHB monitoring [26,27,28]. Several studies have suggested that limited English proficiency, not having a regular source of care, and lack of health insurance coverage are among the major barriers that prevent Asian Americans from receiving regular CHB monitoring [28,29,30,31,32,33].

Research has also found that low knowledge of and lack of awareness of hepatitis and liver cancer risk is another significant predictor of poor monitoring in various Asian American subgroups [33,34,35,36]. In addition, psychosocial factors, such as self-efficacy, and motivation have been suggested as predictors of adherence to CHB monitoring [29,37,38], yet empirical evidence was lacking. In recent years, a theoretical model known as the information-motivation-behavioral skills (IMB) model has been used to better explain factors that influence healthy behavior [39]. The IMB model draws from rational behavior theory and social cognitive theory to propose that healthy behaviors are based on three components: information, motivation, and behavioral skills. Information refers to an accurate knowledge of the specific healthy behavior. Motivation refers to personal and social motivation to perform a specific behavior. Behavioral skills include an individual’s self-efficacy and confidence in performing the behavior [40]. While the IMB model has been widely used in research on the prevention and management of HIV/AIDS [41,42,43] and diabetes [44,45], it has not been applied to adherence to CHB monitoring, to the best of our knowledge.

Guided by the IMB model, we examined factors associated with CHB monitoring adherence among Asian Americans living with CHB. Our findings advance the understanding of relationships among cognitive, emotional, and behavioral skills factors that influence CHB monitoring behaviors in the IMB model and have critical implications for the development and implementation of evidence-based interventions for CHB monitoring and liver cancer prevention in the Asian American community.

## 2. Materials and Methods

### 2.1. Study Participants

From March 2019 to March 2020, 382 Asian American CHB patients, including 298 Chinese Americans and 84 Vietnamese Americans, were enrolled through combined recruitment approaches in the Greater Philadelphia Area and New York City to participate in a randomized control trial on long-term CHB monitoring and antiviral treatment adherence. The eligibility criteria for new subjects were: (1) aged 18 or above; (2) self-identified Chinese or Vietnamese ethnicity; (3) CHB infection with positive HBV surface antigen (HBsAg); (4) CHB diagnosed for at least 12 months; (5) non-compliance with HBV monitoring and treatment guidelines for more than 6 months; (6) cell phone accessible by receiving text messaging; (7) not enrolled in any other HBV Management intervention (to prevent a potential program impact). We excluded cases with missing data on HBV monitoring behaviors. The analysis sample consisted of 378 participants, including 295 Chinese and 83 Vietnamese Americans.

### 2.2. Procedures

The current study used the data collected from the baseline survey of the larger trial. Baseline data collection was conducted from April 2019 to March 2020 through a face-to-face survey, which took approximately 30 min to complete. The survey questionnaire was developed in English, translated into Mandarin, Cantonese, and Vietnamese, and back-translated into English by bilingual community health educators on the research team. The back-translated version was compared with the English version to verify that the questions were properly translated. Participants completed the survey in their preferred language. During the survey, trained bilingual community health educators provided language support to address questions from the participants. The study was approved by the Western Institutional Review Board (WIRB) (protocol #: 20190122). All participants read and signed informed consent forms to participate in the study and received a $25 incentive for completing the baseline survey.

### 2.3. Measures

The primary outcomes were HBV monitoring measures, including doctor visits for CHB and HBV blood testing in the past 6 months. Doctor visits for HBV were measured with one question “Did you see a doctor to check your hepatitis B infection condition during the past six months?” The answers were dichotomous (yes or no). HBV blood testing was measured with the question “Have you had blood tests (e.g., HBV DNA, Liver function, Alpha-fetoprotein) in the past six months?” The answers were dichotomous (yes or no). For both questions, “I don’t know” was coded as “no”.

HBV-related knowledge was examined with a 10-item scale that previously was validated in Asian Americans with CHB [46]. Specifically, participants answered “False”, “True”, or “Don’t know” to 10 HBV knowledge statements, such as “People will feel sick if they are infected with hepatitis B” and “Regular monitoring and treatment can reduce liver damage caused by chronic hepatitis B”. One point was assigned to correct answers and zero points to wrong answers. We computed the accumulated HBV knowledge score by summing the points from all 10 items. The final knowledge score ranged from 0 to 10, with a higher numeric value indicating a higher level of HBV-related knowledge.

CHB-Management motivation was accessed with 10 items on a five-point Likert scale (from 1, “strongly agree”, to 5, “strongly disagree”). Examples of the items include “I don’t like taking my HBV medications because they remind me that I am HBV+” and “My healthcare provider doesn’t give me enough support when it comes to taking my medications as prescribed”. The total motivation score was the summation of the responses to the 10 items. The score ranged from 5 to 50, with a higher numeric value indicating a higher level of motivation related to CHB management.

CHB-Management self-efficacy was measured with 13 items adapted from the medical adherence self-efficacy score (MASES) [47], asking participants how confident they were in taking HBV medications under various situations. The answers ranged from 0 “not at all” to 3 “extremely sure”, which were summed up to compute the final self-efficacy score. The self-efficacy score ranged from 0 to 26, with a higher numeric value indicating a greater confidence in taking HBV medications as recommended by doctors.

Socioeconomic factors, specifically, participants’ age in years, gender, ethnicity, U.S. residency length, marital status, education levels, employment status, annual household income, health insurance coverage, and English-speaking proficiency were included as predictors of HBV management outcomes.

### 2.4. Statistical Analysis

We conducted a descriptive analysis of the sociodemographic characteristics and the psychosocial factors of the analysis sample. We also conducted chi-square tests and *t*-tests to examine associations between sociodemographic/psychosocial factors and CHB management behaviors (doctor’s visit and blood test). We then fitted two multilevel mixed-effects generalized linear models (GLMs) to identify the significant predictors of the outcomes while accounting for the sample clustering by recruitment site and state. All data analyses were conducted using Stata 16 [48]. A *p* value that was smaller than 0.05 was considered statistically significant.

## 3. Results

Table 1 presents the sociodemographic characteristics and psychosocial factors of the participants, as well as their association with CHB monitoring, specifically doctor’s visit in the past six months. Bivariate analyses showed that being Chinese (vs. Vietnamese), having a high school or lower education (vs. college or above), having health insurance, and having a regular physician were significantly associated with higher rates of having visited a doctor for CHB in the past six months. In addition, having a higher HBV-related knowledge score, a higher CHB-management motivation score, and a higher CHB-managed self-efficacy were significantly associated with higher rates of having visited a doctor for CHB in the past six months.

Table 2 presents the sociodemographic characteristics and psychosocial factors of the participants, as well as their association with having had blood tests for their CHB infection in the past six months. Bivariate analyses showed similar associations with those in Table 1. Specifically, being Chinese (vs. Vietnamese), having a high school or lower education (vs. college or above), having health insurance, and having a regular physician were significantly associated with higher rates of having had blood tested for CHB in the past six months. In addition, having a higher HBV-related knowledge score, a higher CHB-management motivation score, and a higher CHB-managed self-efficacy were significantly associated with higher rates of blood testing for CHB in the past six months. Other characteristics, including household income, were not significantly associated with the outcome.

Table 3 presents the results of the multivariate logistic regression on a doctor’s visit and blood testing in the past six months. Having a regular physician (OR = 3.81, *p* < 0.05) was significantly associated with a higher likelihood of having visited a doctor for CHB, while Chinese ethnicity (vs Vietnamese), having a high school or lower degree (vs. college degree or above), and having health insurance were significant predictors of having had blood tests done in the past six months. With regard to psychosocial factors, higher HBV-related knowledge and higher CHB-management motivation were significantly associated with a higher likelihood of both outcomes—having had a doctor’s visit and blood tests in the past six months. CHB-management self-efficacy was not a significant predictor for either outcome.

## 4. Discussion

The present study examined factors associated with CHB monitoring adherence among Chinese and Vietnamese Americans living with CHB. More specifically, we examined adherence within the framework of the IMB model to better understand the impact of cognitive, emotional, and behavioral factors on the pursuit of healthy behaviors. The results of our analyses indicate that HBV knowledge and CHB-management motivation are significant predictors of having a doctor’s visit in the past six months for both Chinese and Vietnamese American study participants. We further found that HBV knowledge and CHB-management motivation are positively associated with the likelihood of having had blood tests for HBV in the past 6 months in these populations.

Previous studies have found that low levels of knowledge and lack of awareness about hepatitis and the risk of liver cancer are important predictors of suboptimal hepatitis monitoring in different subgroups of Asian Americans [33,34,35,36]. Our data support these associations, specifically in that a higher HBV-related knowledge score was significantly associated with higher rates of having visited a doctor for CHB in the past six months and with higher rates of blood testing for CHB in the last six months. Research has also suggested that adherence to CHB monitoring can be predicted by motivation and psychosocial factors (e.g., self-confidence and self-efficacy) [29,37,38]. Our findings also support these observations, with higher CHB-management motivation score and higher CHB-managed self-efficacy being significantly associated with higher rates of having visited a doctor and undergoing blood testing for CHB in the last six months.

In addition, our findings confirmed the significant roles of having health insurance and having a regular physician in CHB monitoring found in previous studies [29,31]. Specifically, we found three-fold differences in the odds of having an office visit and having blood tests done for their CHB conditions by insurance status and whether they had a regular physician. More efforts are needed in outreach, education, and service to Asian Americans without health insurance or a regular source of care.

Household income was not significantly associated with CHB monitoring in bi-variate analysis. Previous research has generated conflicting findings in the relationship between household income level and healthcare utilization among Asian Americans [49,50]. Our findings suggest that individual level education and healthcare access potentially played bigger roles in influencing CHB monitoring behaviors than did household income.

A key strength of the present study is the use of the IMB model, which incorporates components of rational behavior theory and social cognitive theory. The IMB model proposes that engaging in a healthy behavior requires accurate knowledge of the behavior, personal and social motivation to perform the behavior, and individual self-efficacy and confidence to execute the behavior [40]. To the best of our knowledge, the present study is the first to apply the IMB model to CHB monitoring adherence. Another major strength of our study is the baseline data, which was drawn from a randomized control trial on long-term CHB monitoring and antiviral treatment adherence in a unique study population, encompassing Chinese and Vietnamese Americans, from the Greater Philadelphia Area and New York City.

The findings of this study serve as important baseline data for the large-scale randomized control trial intervention. The culturally tailored, multilevel components that we have designed and implemented are aimed to address the barriers that Asian Americans with CHB experience on the healthcare system level, provider level, community level, and individual level. The multilevel approach is critical in the empowerment of this vulnerable population, especially in the contexts of structural racism, anti-Asian discrimination, and other difficulties exacerbated by the COVID-19 pandemic [51,52]. More efforts are needed in creating structural level interventions to facilitate policy and systemic change to improve access to care for the underprivileged and medically underserved populations [52].

## 5. Conclusions

In conclusion, we examined factors associated with CHB monitoring adherence among Asian Americans living with CHB, using the IMB model as a guide. Our primary finding, that higher HBV-related knowledge and greater CHB-management motivation are significantly associated with carrying out CHB-monitoring behaviors, supports observations from previous studies and offers new insight into cognitive, emotional, and behavioral skills factors that influence healthy behaviors among Chinese and Vietnamese Americans. Moreover, our findings could have important implications for the development of novel evidence-based interventions for CHB monitoring and liver cancer prevention in the Asian American community.

## Figures and Tables

**Table 1 ijerph-19-04642-t001:** Sociodemographic and psychosocial factors among participants who had visited doctors in the past six months for their CHB and those who did not.

	Visited Doctor for CHB (*n* = 132)	No Visit (*n* = 246)	$\chi^{2}\;\left(df\right)\;or\;t\;\left(df\right),\;p$
*Sociodemographic characteristics*	*n* (%) or mean (sd)		
Age	54.26 (13.92)	52.75 (13.30)	−1.03 (376), 0.30
Gender			0.49 (1), 0.48
Female	71 (36.60%%)	123 (63.40%)	
Male	61 (33.15%)	123 (66.85%)	
Ethnicity			9.76 (1), 0.002
Chinese	115 (38.98%)	180 (61.02%)	
Vietnamese	17 (20.48%)	66 (79.52%)	
Marital status			1.95 (1), 0.16
Currently married/cohabitating	111 (36.39%)	194 (63.61%)	
Other	19 (27.54%)	50 (72.46%)	
Born in the US			0.50 (1), 0.48
No	131 (35.12%)	242 (64.88%)	
Yes	1 (20.00%)	4 (80.00%)	
Years lived in the US			1.50 (1), 0.22
<10 years	13 (27.66%)	34 (72.34%)	
≥10 years	116 (36.83%)	199 (63.17%)	
Educational attainment			7.25 (1), 0.01
≤high school	103 (39.31%)	159 (60.69%)	
≥college	29 (25.00%)	87 (75.00%)	
Employment status			1.58 (2), 0.45
Employed	78 (33.33%)	156 (66.67%)	
Unemployed	9 (32.14%)	19 (67.86%)	
Not in labor force	44 (40.00%)	66 (60.00%)	
Annual household income			<0.001 (1), 0.99
0–$19,999	67 (34.90%)	125 (65.10%)	
≥$20,000	65 (34.95%)	121 (65.05%)	
Having health insurance			12.15 (1), <0.001
No	8 (14.29%)	48 (85.71%)	
Yes	123 (38.32%)	198 (61.68%)	
Having a regular physician			12.83 (1), <0.001
No	6 (12.24%)	43 (87.76%)	
Yes	116 (38.54%)	185 (61.46%)	
*Psychosocial factors*			
HBV-related knowledge	5.92 (2.11)	5.29 (2.54)	−2.38 (360), 0.02
CHB management motivation	28.80 (7.04)	26.02 (6.73)	−3.69 (359), <0.001
CHB management self-efficacy	6.02 (6.41)	3.62 (6.42)	−3.36 (353), <0.001

**Table 2 ijerph-19-04642-t002:** Sociodemographic and psychosocial factors among participants who had blood tests done in the past six months and those who did not.

	Had Blood Tests Done (*n* = 177)	No Tests (*n* = 188)	$\chi^{2}\;\left(df\right)or\;t\;\left(df\right),\;p$
*Sociodemographic characteristics*	*n* (%) or mean (sd)		
Age	52.66 (13.59)	53.97 (13.32)	0.93 (363), 0.35
Gender			0.34 (1), 0.56
Female	93 (50.00%)	93 (50.00%)	
Male	84 (46.93%)	95 (53.07%)	
Ethnicity			40.60 (1), <0.001
Chinese	163 (57.39%)	121 (42.61%)	
Vietnamese	14 (17.28%)	67 (82.72%)	
Marital status			1.72 (1), 0.19
Currently married/cohabitating	147 (50.00%)	147 (50.00%)	
Other	28 (41.18%)	40 (58.82%)	
Born in the US			1.65 (1), 0.20
No	176 (48.89%)	184 (51.11%)	
Yes	1 (20.00%)	4 (80.00%)	
Years lived in the US			3.37 (1), 0.07
<10 years	17 (36.96%)	29 (63.04%)	
≥10 years	156 (51.49%)	147 (48.51%)	
Educational attainment			7.82 (1), 0.01
≤high school	135 (53.36%)	118 (46.64%)	
≥college	42 (37.50%)	70 (62.50%)	
Employment status			5.54 (2), 0.06
Employed	105 (46.67%)	120 (53.33%)	
Unemployed	9 (33.33%)	18 (66.67%)	
Not in labor force	61 (56.48%)	47 (43.52%)	
Annual household income			.02 (1), 0.89
0–$19,999	90 (48.13%)	97 (51.87%)	
≥$20,000	87 (48.88%)	91 (51.12%)	
Having health insurance			10.75 (1), 0.001
No	15 (27.78%)	39 (72.22%)	
Yes	161 (51.94%)	149 (48.06%)	
Having a regular physician			17.11 (1), <0.001
No	10 (21.28%)	37 (78.72%)	
Yes	156 (53.79%)	134 (46.21%)	
*Psychosocial factors*			
HBV-related knowledge	5.76 (2.04)	5.24 (2.70)	−2.02 (348), 0.04
CHB management motivation	29.47 (7.37)	24.77 (5.83)	−6.64 (347), <0.001
CHB management self-efficacy	6.44 (7.12)	2.70 (5.40)	−5.50 (341), <0.001

**Table 3 ijerph-19-04642-t003:** Multivariate logistic regression results on doctor visit and blood test.

	Had Visited Doctor for CHB	Had Blood Tests Done
*predictors*	OR (95% CI)	
Female gender (ref: male)	1.23 (0.68–2.23)	1.44 (.74–2.78)
Vietnamese ethnicity (ref: Chinese)	0.62 (0.22–1.73)	0.03 (0.01–10) ***
College degree or above (ref: <= high school)	0.50 (0.22–1.12)	0.37 (0.15–90) *
Lived in the US for 10+ years (ref: < 10 yrs)	1.03 (0.40–2.65)	1.35 (0.51–3.61)
Having health insurance (ref: no)	2.99 (0.67–5.93)	4.34 (1.24–15.27) *
Having a regular physician (ref: no)	3.81 (1.21–12.01) *	2.91 (0.86–9.89)
Speaking English well/very well (ref: no/poor)	0.87 (0.40–1.90)	1.41 (0.60–3.32)
HBV-related knowledge	1.24 (1.07–1.45) **	1.29 (1.10–1.51) **
CHB-management motivation	1.06 (1.01–1.11) *	1.12 (1.06–1.19) ***
CHB-management self-efficacy	1.03 (0.98–1.09)	1.02 (0.96–1.09)
Constant	0.004 ***	0.001 ***
n	313	301
Log likelihood	−144.63	−126.59
$\mathrm{Wald}\;\chi^{2}$ (df), *p*	33.77 (10), <0.001	64.89 (10), <0.001

* *p* < 0.05; ** *p* < 0.01; *** *p* < 0.001. Abbreviations: OR = odds ratio; CI = confidence interval.

## Data Availability

The data presented in this study are available on request from the corresponding author.

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
