# Peer review of "Examining the Influencing Factors of Chronic Hepatitis B Monitoring Behaviors among Asian Americans: Application of the Information-Motivation-Behavioral Model"

_ijerph, 2022, doi:10.3390/ijerph19084642_

Round 1

Reviewer 1 Report

Major concerns:

  1. Page 3. lines 99-112 report the rules for compiling materials and methods section and not the actual datasets used, ethics committee approval number... and should be changed with the relevant information
  2. From Table 2 it appears that household income is not a predictive factor for CHB monitoring adherence and/or testing. If the data supports this conclusion, it should be highlighted in the discussion
  3. Discussion could be improved with a description of relevant differences, to improve readability

Minor concerns:

  1. Cirrhosis is by definition a liver disease, so the term "liver cirrhosis" (page 2, line 47) is redundant

Author Response

Reviewer 1 Comments

Response

1.       Page 3. lines 99-112 report the rules for compiling materials and methods section and not the actual datasets used, ethics committee approval number... and should be changed with the relevant information

Removed

2.       From Table 2 it appears that household income is not a predictive factor for CHB monitoring adherence and/or testing. If the data supports this conclusion, it should be highlighted in the discussion

We added the content in the discussion section: Household income was not significantly associated with CHB monitoring in bi-variate analysis. Previous research has generated conflicting findings in the relationship between household income level and healthcare utilization among Asian Americans. Our findings suggest that individual level education and healthcare access potentially played bigger roles in influencing CHB monitoring behaviors than did household income.

3.       Discussion could be improved with a description of relevant differences, to improve readability

We added content in the discussion section on the differences in outcomes by some of the significant predictors.

4.       Cirrhosis is by definition a liver disease, so the term "liver cirrhosis" (page 2, line 47) is redundant

We removed the word “liver”

Reviewer 2 Report

The manuscript by Grace et al shows that higher HBV-related knowledge and greater chronic hepatitis B (CHB) management motivation are significantly associated with carrying out CHB-monitoring behaviors.

I have the following comments: 

  1. What is the impact of household income and having social security on your conclusions? How do your data compare to studies performed in EU countries?
  2.  lines 99 to 112. This information (for authors) should be removed.

Author Response

Reviewer 2 Comments

Response

1.       What is the impact of household income and having social security on your conclusions? How do your data compare to studies performed in EU countries?

We discussed the role of household income in the discussion section. While we did examine the roles of having health insurance in influencing CHB monitoring behavior, our survey did not ask if they have “social security benefit.” We will consider including this factor in future research.

While we could not find any published articles on CHB monitoring in Asians in EU countries, we expect that the differences in healthcare system, health insurance policies, and the immigration profiles would impact the differential CHB monitoring behaviors between Asians in the EU countries and Asians in the US.

2.       lines 99 to 112. This information (for authors) should be removed.

Removed